# A Role for Human Renal Tubular Epithelial Cells in Direct Allo-Recognition by CD4+ T-Cells and the Effect of Ischemia-Reperfusion

**DOI:** 10.3390/ijms22041733

**Published:** 2021-02-09

**Authors:** Theodoros Eleftheriadis, Georgios Pissas, Marta Crespo, Evdokia Nikolaou, Vassilios Liakopoulos, Ioannis Stefanidis

**Affiliations:** 1Department of Nephrology, Faculty of Medicine, University of Thessaly, 41110 Larissa, Greece; gpissas@msn.com (G.P.); nikolaoueyh@gmail.com (E.N.); liakopul@otenet.gr (V.L.); stefanid@med.uth.gr (I.S.); 2Nephrology Department, Institut Hospital del Mar d’Investigacions Mèdiques, Hospital del Mar, Parc de Salut Mar, 08003 Barcelona, Spain; mcrespo@parcdesalutmar.cat

**Keywords:** kidney transplantation, renal tubular epithelial cells, direct allorecognition, CD4+ T-cells, ischemia-reperfusion, rejection

## Abstract

Direct allorecognition is the earliest and most potent immune response against a kidney allograft. Currently, it is thought that passenger donor professional antigen-presenting cells (APCs) are responsible. Further, many studies support that graft ischemia-reperfusion injury increases the probability of acute rejection. We evaluated the possible role of primary human proximal renal tubular epithelial cells (RPTECs) in direct allorecognition by CD4+ T-cells and the effect of anoxia-reoxygenation. In cell culture, we detected that RPTECs express all the required molecules for CD4+ T-cell activation (HLA-DR, CD80, and ICAM-1). Anoxia-reoxygenation decreased HLA-DR and CD80 but increased ICAM-1. Following this, RPTECs were co-cultured with alloreactive CD4+ T-cells. In T-cells, zeta chain phosphorylation and c-Myc increased, indicating activation of T-cell receptor and co-stimulation signal transduction pathways, respectively. T-cell proliferation assessed with bromodeoxyuridine assay and with the marker Ki-67 increased. Previous culture of RPTECs under anoxia raised all the above parameters in T-cells. FOXP3 remained unaffected in all cases, signifying that proliferating T-cells were not differentiated towards a regulatory phenotype. Our results support that direct allorecognition may be mediated by RPTECs even in the absence of donor-derived professional APCs. Also, ischemia-reperfusion injury of the graft may enhance the above capacity of RPTECs, increasing the possibility of acute rejection.

## 1. Introduction

Kidney transplantation is the best treatment option for patients with end-stage kidney disease. However, despite the application of immunosuppressive protocols with severe side-effects, long-term graft survival remains relatively low, mainly due to antibody-mediated rejection [1,2].

Moreover, the shortage of kidney transplants led to the use of non-optimal grafts, which inevitably increases the incidence of the ischemia-reperfusion injury-induced delayed graft function [3]. According to many [4,5,6,7,8], but not all studies [9,10,11], delayed graft function is associated with stimulation of the host immune system and a higher incidence of acute rejection. It has been suggested that kidney transplant ischemia-reperfusion injury induces cell necrosis and release of damage-associated molecular patterns. The latter are recognized by specific receptors, such as the Toll-like receptors, leading to activation of innate immune cells and the production of cytokines and chemokines. This inflammatory milieu induces further renal cell necrosis through necroptosis and in parallel recruits, and activates cells of the adaptive immune system facilitating graft rejection [12].

Thus, studying the mechanisms that govern the alloimmune response is pivotal for planning new therapeutic strategies and improving graft survival. At present, three T-cell allorecognition pathways have been identified [13,14,15,16]. It should be noted that T-cell allorecognition besides cellular rejection determines antibody-mediated rejection as well, due to the central role of CD4+ T-cells in adaptive immunity [17]. Accordingly, in a mouse kidney transplantation model, depletion of CD4+ T-cells eliminated donor-specific alloantibodies [18].

In the direct T-cell allorecognition pathway [13,14,15,16], recipient CD4+ and CD8+ T-cells recognize intact Major Histocompatibility Complex (MHC) Class II and MHC Class I molecules, respectively, on donor professional antigen-presenting cells (APCs), mainly dendritic cells, that are transferred with the graft. Indirect allorecognition is a form of conventional antigen presentation. In the indirect T-cell allorecognition pathway [13,14,15,16], graft antigens, mostly MHC molecules, are internalized, processed, and presented on MHC molecules by recipient professional APCs to recipient T-cells. Finally, in the semi-direct T-cell allorecognition pathway [16], graft MHC molecules are acquired by recipient dendritic cells and presented intact to recipient T-cells. The exact role of the semi-direct pathway remains to be elucidated [16]. The indirect pathway can become active at any time and has been incriminated for late graft cellular and humoral rejection [13,14,15,16]. The direct pathway elicits the most potent immune response since up to 10% of recipient T-cells recognize a single MHC alloantigen. Also, every MHC molecule on the surface of professional donor APCs is identified as foreign by the specific recipient T-cells. On the contrary, in conventional processing and presentation of an antigen, only around 150 MHC-antigen-derived peptide complexes on the surface of an APC are recognized by the specific host T-cells. Currently, it is thought that the direct pathway plays a role during the immediate post-transplantation period since donor-derived dendritic cells are eliminated by the recipient immune system shortly after transplantation [13,14,15,16].

Hence, according to the generally accepted T-cell allorecognition pathways, decreasing alloreactivity as time passes from transplantation may result from the elimination of donor professional APCs. However, renal tubular epithelial cells have all the means required for effective antigen presentation to CD4+ T-cells [19,20,21,22]. They display on their surface the necessary for the CD4+ T-cell activation MHC Class II and co-stimulatory molecules, as well as the cell adhesion molecules, which are required for an effective immunological synapse formation between the renal tubular epithelial cell and the CD4+T-cell [23,24]. Remarkably, tubulitis characterizes the renal biopsies from patients with acute cellular rejection. In tubulitis, mononuclear cells, including CD4+ T-cells, are present inside the tubular basement membrane and in attachment with renal tubular epithelial cells [25,26,27].

Notably, several types of epithelial cells have been found to present antigen to CD4+ T-cells [28,29,30,31,32]. Interestingly, mouse renal proximal tubular epithelial cells (RPTECs) have the capacity to present a protein antigen to CD4+ T-cells and induce their activation and differentiation towards an inflammatory phenotype [32]. Thus, the ability of renal tubular epithelial cells to drive a potent and constant direct alloreactive immune response cannot be excluded.

In this study, we evaluated whether human RPTECs can be subjected to direct allorecognition by CD4+ T-cells. If this is the case, then the direct allorecognition inducing graft cells are many more than is currently thought. Also, since no short-lived donor APCs are required, direct allorecognition would last as long as the graft remains implanted. Finally, because of supporting evidence for an association between ischemia-reperfusion injury of the kidney graft and subsequent acute rejection [4,5,6,7,8], we assessed whether ischemia-reperfusion affects the ability of RPTECs to activate alloreactive CD4+ T-cells.

## 2. Results

### 2.1. RPTECs Express All the Required Molecules for Direct Allorecognition by CD4+ T-Cells and Produce MCP-1, whereas Anoxia-Reoxygenation Affects All the Above

Firstly, we evaluated whether RPTECs express the molecules to form an effective immune synapse with CD4+ T-cells. More precisely, we assessed the existence of MHC class II molecules, represented by the HLA-DR, of co-stimulatory molecules, represented by the CD80, and of the ICAM-1.

Indeed, RPTECs expressed all the above molecules. Anoxia-reoxygenation altered their levels significantly. It decreased HLA-DR by a factor of 0.62 ± 0.04 and CD80 by a factor of 0.50 ± 0.10 (Figure 1A,C). On the contrary, anoxia-reoxygenation increased ICAM-1 to 2.20 ± 0.12 times of the control (Figure 1A,D).

Further, in the supernatants of the RPTEC cultures, anoxia-reoxygenation increased the concentration of the chemokine MCP-1 significantly, from 21.6 ± 1.3 pg/mL to 84.8 ± 8.6 pg/mL (Figure 1E).

### 2.2. RPTECs Activate the T-Cell Receptor and the Co-Stimulation Signal Transduction Pathways in CD4+ T-Cells, and Anoxia-Reoxygenation Intensifies the above Signal Transduction Pathways Further

In CD4+ T-cells collected after 4 h of RPTEC CD4+ T-cell co-culture, p-zeta increased significantly to 1.7 ± 0.15 of the control, demonstrating activation of the T-cell receptor (TCR) signal transduction pathway. When RPTECs were previously subjected to anoxia, p-zeta level was increased further to 3.05 ± 0.47 times of the control (Figure 2A,B). Zeta-chain remained unaffected since its expression in CD4+ T-cells harvested from RPTEC CD4+ T-cell co-culture was 1.07 ± 0.06 times the control, and in the case of previous RPTEC culture under anoxia 1.12 ± 0.11 times the control (Figure 2A,C).

In CD4+ T-cells collected after 4 h of RPTEC CD4+ T-cell co-culture, c-Myc expression raised significantly to 2.04 ± 0.18 of the control, indicating activation of the co-stimulation signal transduction pathway. When RPTECs were previously subjected to anoxia, c-Myc was enhanced further to 3.66 ± 0.46 times of the control (Figure 2A,D).

### 2.3. RPTECs Trigger CD4+ T-Cells to Proliferate, Anoxia-Reoxygenation Further Increases Proliferation, whereas CD4+ T-Cell Differentiation to Regulatory T-Cells Does Not Occur

BrdU assay showed that, after 7 days of RPTEC CD4+ T-cell co-culture, and compared to isolated CD4+ T-cell, proliferation index increased significantly from 1.0 to 1.90 ± 0.75. In the case of previous RPTEC culture under anoxia, the proliferation index rose further to 2.83 ± 0.88 (Figure 3A).

The proliferation of CD4+ T-cells was also confirmed by the expression of the proliferation marker Ki-67 in CD4+ T-cells collected after 3 days of RPTEC CD4+ T-cell co-culture. Compared to isolated CD4+ T-cell culture, RPTECs increased Ki-67 expression in CD4+ T-cells significantly 1.91 ± 0.30 times. When RPTECs were previously subjected to anoxia, Ki-67 rose further to 2.86 ± 0.46 times of the control (Figure 3B,C). As depicted in Figure 3B, both Ki-67 alternatively spliced variants with an approximate molecular weight of 395 kDa and 345 kDa were detected by Western blotting [33].

Compared to isolated CD4+ T-cell culture, in CD4+ T-cells harvested after 3 days of RPTEC CD4+ T-cells co-culture, the signature regulatory T-cell (Treg) transcription factor FOXP3 did not change (0.99 ± 0.08 of the control). When RPTECs were previously subjected to anoxia, FOXP3 remained unaffected as well (1.02 ± 0.09 of the control) (Figure 3B,D).

## 3. Discussion

Despite the administration of potent immunosuppressive regimens, long-term graft survival remains relatively low. Today the leading cause of death-censored graft failure is rejection, characterized mainly as antibody-mediated rejection [1,2]. Additionally, patient’s death with a functioning graft, for reasons that frequently are associated with immunosuppressive medications, is still the most common cause of graft loss [34]. Thus, delineating the exact molecular mechanisms that govern transplantation immunology is pivotal since it may lead to more effective immunomodulatory strategies.

Direct allorecognition is the earliest and the most potent de novo donor immune response against the kidney transplant. It is thought that direct allorecognition plays a significant role during the early post-transplantation period since it is dependent on donor passenger professional APCs within the graft. The latter are eventually eliminated by the host immune system [13,14,15,16]. In this study, firstly, we evaluated the ability of renal tubular epithelial cells to mediate direct allorecognition.

Like other studies [19,20,21,22], we noticed that RPTECs display all the necessary components for CD4+ T-cell activation [17,23]. More precisely, we detected in RPTECs MHC Class II molecules, assessed in our research by HAL-DR, and co-stimulatory molecules, evaluated in our study by CD80. Interestingly, co-stimulatory molecules may serve as diagnostic and prognostic markers in various kidney diseases, indicating a possible role of T-cells in their pathogenesis [35]. Cell-adhesion molecules have been studied in nephrology, mainly in the context of atherosclerosis that frequently accompanies chronic kidney disease [36]. However, the cell adhesion molecule ICAM-1 is also necessary for an effective immunological synapse between the APC and the T-cell [23]. We detected the presence of ICAM-1 in RPTECs.

Previous studies have shown that various types of epithelial cells, such as upper airway [30], intestinal [28,29], liver [31], and renal tubular epithelial cells [32], which are not professional APCs, can process foreign protein antigen and present its fragments along with their-own MHC Class II molecules to specific CD4+ T-cells. Direct allorecognition is much more potent than the recognition of a conventionally presented antigen since up to 10% of T-cells recognize a single MHC alloantigen [13,14,15,16]. In contrast, only a small proportion of the T-cell population responds to a specific antigen [approximately 1 cell per 10^5^ to 10^6^ T-cells] [13,14,15,16]. Thus, the hypothesis that renal epithelial cells may induce direct allorecognition seems rational.

To test the above hypothesis, we co-cultured RPTECs with CD4+ T-cells derived from random individuals. Compared to isolated CD4+ T-cells, in CD4+ T-cells derived from RPTEC CD4+ T-cell co-culture, p-zeta-chain levels were increased. Zeta-chain phosphorylation is a very proximal event after the MHC Class II TCR engagement. It is the first covalent chemical bond formed in the TCR signal transduction pathway [37,38]. Hence, our result indicates that RPTECs activate the TCR of alloreactive CD4+ T-cells.

Also, we showed that RPTECs enhance c-Myc expression in CD4+ T-cells. Co-stimulation, and especially CD80/86 interaction with CD28, is necessary for c-Myc upregulation in T-cells, which in turn, by inducing aerobic glycolysis, favors T-cell proliferation [39,40,41,42,43]. Remarkably, a recent study showed that in human CD4+ T cells, CD28 upregulates glycolysis independently of TCR engagement by increasing c-Myc [39]. Thus, our data indicate that RPTECs provide co-stimulatory signals to alloreactive CD4+ T-cells.

Moreover, using two different approaches, the BrdU assay in RPTEC CD4+ T-cell co-culture, and the expression of the proliferation marker Ki-67 in CD4+ T-cells collected from RPTEC CD4+ T-cell co-culture [44], we showed that RPTEC-induced TCR and co-stimulation activation eventually promotes alloreactive CD4+ T-cell clonal expansion. Due to the central role of CD4+ T-cell in the adaptive immune response, the ability of RPTECs to activate alloreactive CD4+ T-cells should affect all the components of the immune response against the kidney transplant [17]. Likewise, in a mouse kidney transplantation model, depletion of CD4+ T-cells eliminated donor-specific alloantibodies [18].

Conventional antigen presentation does not always result in CD4+ T-cell differentiation towards an effector phenotype. For instance, conventional antigen processing and presentation of ovalbumin by mouse RPTECs induces an inflammatory phenotype in specific CD4+ T-cells [32]. On the contrary, antigen presentation by hepatocellular cells induces a regulatory phenotype in CD4+ T-cells [31]. The latter may contribute to the known immunosuppressive state in the case of simultaneous liver and kidney transplantation [45]. To test which of the above occurs in our alloreactivity model, we evaluated the expression of FOXP3 in CD4+ T-cells harvested from RPTEC CD4+ T-cell co-culture. We found that the presence of RPRECs does not affect FOXP3 level in alloreactive CD4+ T-cells. Since FOXP3 is the signature transcription factor of Treg [46], our results indicate that the proliferating CD4+ T-cells are not differentiated into Treg.

Delayed graft function, the clinical consequence of peri-transplantation period kidney graft ischemia-reperfusion injury, has been associated by many [4,5,6,7,8], albeit not all studies [9,10,11], with acute rejection episodes. To evaluate whether ischemia-reperfusion affects the ability of RPTECs to activate alloreactive CD4+ T-cells directly, we performed the above described RPTEC CD4+ T-cell co-cultures with RPTECs that have been previously subjected to anoxia. We found that compared to untreated RPTECs, anoxia-subjected RPTECs trigger more potently the TCR and the co-stimulatory signal transduction pathways in alloreactive CD4+ T-cells. Also, they stimulated more the CD4+ T-cell proliferation without promoting their differentiation into Treg. Therefore, the increased ability of the subjected to anoxia-reoxygenation RPTECs to evoke an alloreactive CD4+ T-cell immune response may contribute to the observed association between delayed graft function and acute rejection.

We also assessed the effect of anoxia-reoxygenation on the RPTEC molecules required for the CD4+ T-cell activation. In RPTECs, anoxia-reoxygenation decreased HLA-DR and CD80 expression, but on the other hand, it enhanced ICAM-1 expression. The increased capacity of anoxia-subjected RPTECs to stimulate alloreactive CD4+ T-cells could be attributed to the latter. ICAM-1 is necessary for immunological synapse formation with CD4+ T-cells [23]. Fluorescence video microscopy has shown that during T-cell-B-cell interaction, the last serving as an APC, the initial elevation of intracellular calcium in the T-cell is immediately followed by a rapid ICAM-1 accumulation on the B-cell surface and, more specifically, at the tight interface between the two cells. This increased density of ICAM-1 results in sustained elevation of intracellular calcium in the T-cell, indicating its activation [20]. The role of ICAM-1 in conventional antigen presentation has also been confirmed in experiments with T-cells that lack the ICAM-1 ligand lymphocyte function-associated antigen 1 (LFA-1). In the absence of LFA-1, 100-fold more antigen is required for effective APC-T-cell conjunction and subsequent T-cell activation [47]. Regarding the alloimmunity reactions, mononuclear cells derived from ICAM-1 knock-out mice provide insignificant stimulation in mixed lymphocyte reaction, whereas they proliferate normally as responder cells [48]. Thus, in our experiments, the significantly increased anoxia-reoxygenation-induced ICAM-1 expression seems to overcome the HLA-DR and CD80 reduction as regards the potency of the immunological synapse to activate the alloreactive CD4+ T-cells.

Interestingly, we also detected that anoxia-reoxygenation increased MCP-1 production by the RPTECs. Since the MCP-1 chemokine attracts both monocytes and T-cells [49,50], the above effect of anoxia-reoxygenation may facilitate the interaction between RPTECs and CD4+ T-cells. MCP-1 production by the RPTECs may also contribute to the observed presence of mononuclear cells, including CD4+ T-cells, inside the tubular basement membrane and in attachment with renal tubular epithelial cells in the tubulitis that characterizes kidney biopsies in acute cellular rejection [25,26,27].

Our results indicate that kidney graft antigenicity remains high even after the first post-transplantation period since RPTECs are subjected to direct allorecognition. Thus, the search for the kidney transplantation holy grail, the graft tolerance, may be better focused on the recipient’s immune system. For instance, the development of effective methods for enhancing specific to the graft Treg differentiation would be of great interest [51]. Our data also indicate that, at the immediate post-transplantation period, ischemia-reperfusion injury may increase the antigenicity of the kidney graft. Hence, efforts to alleviate kidney transplant ischemia-reperfusion injury by evolving and applying more efficient preservation methods may be proven beneficial [52]. Finally, our study showed that ischemia-reperfusion increases RPTECs’ antigenicity by enhancing ICAM-1 expression. Research for blocking the interaction between cell adhesion molecules is underway [53], and the effect of such an approach on the field of kidney transplantation deserves evaluation.

Our study limitation lies in its in vitro nature since drawing direct conclusions from in vitro studies to the in vivo model is not always safe. Nevertheless, under the strictly controlled in vitro conditions, we were able for the first time to detect that RPTECs can trigger a direct allorecognition response, which currently is thought to be mediated exclusively by donor-derived passenger professional APCs. Further, we were able to evaluate the effect of anoxia-reoxygenation on the above capacity of RPTECs, excluding the impact of other cell types. Thus, our study could be considered a starting point for further investigation on the role of renal tubular epithelial cells in transplantation immunology.

Another limitation of our study is that we did not perform HLA typing in the RPTECs and in the healthy volunteers who offered the CD4+ T-cells. However, since the IPD-IMGT/HLA Database currently contains 6695 HLA Class II allele sequences [54], the probability of the enrolled in the study healthy volunteers to share the same MHC Class II phenotype as the used human primary RPTECs is negligible.

## 4. Materials and Methods

### 4.1. Cells and Culture Conditions

Human primary RPTECs (cat. no. 4100, ScienCell, Carlsbad, CA, USA) were cultured in Complete Epithelial Cell Medium/w kit (cat. no. M6621, Cell Biologics, Chicago, IL, USA), supplemented with epithelial cell growth supplement, antibiotics, and fetal bovine serum. RPTECs were expanded in 75 cm^2^ flasks, and passage three cells were used for the experiments.

In isolated RPTEC cultures, cells were cultured in 12-well plates at a number of 150,000 cells per well for 16 h before the onset of anoxic conditions. The GasPak EZ Anaerobe Container System with Indicator (cat. no. 26001, BD Biosciences, S. Plainfield, NJ, USA) was used to reduce oxygen levels to less than 1%. Cells within the anaerobe container were cultured for 90 min at 37 °C. These anoxic conditions imitate ischemia. Then, cells were washed, supplemented with fresh culture medium, and placed at 37 °C in a humidified atmosphere containing 5% CO_2_ for another 240 min. These reoxygenation conditions mimic reperfusion. Since primary human RPTECs are vulnerable to ischemia-reperfusion injury [55], cell integrity was inspected and confirmed using an inverted microscope (Axiovert 40C, Carl Zeiss Light Microscopy, Göttingen, Germany) and a digital camera with the related software (3MP USB2.0 Microscope Digital Camera, Amscope, Irvine, CA, USA) (Figure 4A). Five such experiments were performed.

For CD4+ T-cell isolation, blood samples were collected from four healthy volunteers (all men, 35 ± 6 years old). All participants were personnel of our clinic-laboratory. Their medical records were evaluated, and they were subjected to physical examination and routine laboratory tests. Their body mass index (BMI) was within the normal range, and eGFR calculated by the MDRD equation was higher than 90 mL/min/1.73 m^2^. None of them had received a blood transfusion in the past. Informed consent was obtained from the study participants. The Ethics Committee of the University of Thessaly, Faculty of medicine (Larissa, Greece) approved the study protocol (Number of approval:558, date: 10-2-2017).

Ficoll-Hypaque density gradient centrifugation (Histopaque 1077, Sigma-Aldrich; Merck Millipore, Darmstadt, Germany) was used for isolating peripheral blood mononuclear cells (PBMCs) from whole blood. Then, CD4+ T-cells were isolated from PBMCs using the CD4+ T-Cell Isolation Kit, Human (Miltenyi Biotec GmbH, Bergisch Gladbach, Germany). To determine the purity of isolated CD4+ T-cells, we used flow cytometry. Cells were labeled with the FITC Mouse anti-Human CD3 (cat. no. 555339 BD Biosciences, San Jose, CA, USA) and the APC Mouse anti-human CD4 (cat. no. 300514, Biolegend, San Diego, CA, USA). FITC Mouse IgG1, κ Isotype (cat. no. 400110, Biolegend) and APC Mouse IgG1, κ Isotype (cat. no. 400122, Biolegend) were used as isotypic controls. Flow cytometry revealed that more than 98% of isolated cells were CD4+ T-cells (Figure 4B). CD4+ T-cells were counted by optical microscopy on a Neubauer plaque, and cell viability was assessed by trypan blue assay (Sigma-Aldrich).

For RPTEC CD4+ T-cell co-culture, RPTECs cultured in Complete Epithelial Cell Medium were seeded in 24-well plates at a number of 50,000 cells per well or in 96-well plates at a number of 10,000 cells per well. RPTECs were left for 16 h to adhere and then subjected or not to 90 min of anoxia as described above. Then, RPTECs were washed, and CD4+ T-cells were added at a number of 250,000 per well in 24-well plates or 50,000 cells per well in 96-well plates. Afterward, cells were cultured in RPMI 1640 medium with L-glutamine and 10mM 4-(2-hydroxyethyl)-1-piperazineethanesulfonic acid (HEPES) and supplemented with 10% fetal bovine serum (Sigma-Aldrich; Merck Millipore) and antibiotic-antimycotic solution (Sigma-Aldrich; Merck Millipore). RPTEC CD4+ T-cell co-culture was performed at 37 °C in a humidified atmosphere containing 5% CO_2_ for 4 h or 3 days in the 24-well plates or 7 days in the 96-well plates. Thus, in these experiments, in RPTECs previously subjected to anoxia, reoxygenation started along with the addition of CD4+ T-cells. Four such experiments were performed. In order to obtain reliable results, in each of the four RPTEC CD4+ T-cell co-culture experiments, CD4+ T-cells from the same individual were used.

### 4.2. Assessment of the Proteins of Interest

In isolated RPTEC cultures, the adherent RPTECs were lysed after the 240 min reoxygenation phase. Five such experiments were performed. In RPTEC CD4+ T-cell co-cultures, after 4 h or three days of culture, the supernatant containing the CD4+ T-cells was collected in Eppendorf tubes, centrifuged, and the cell pellet was lysed. Four such experiments were performed for each of the two-time points.

The T-PER tissue protein extraction reagent (Thermo Fisher Scientific Inc., Rockford, IL, USA) supplemented with protease and phosphatase inhibitors (Sigma-Aldrich; Merck Millipore and Roche Diagnostics, Indianapolis, IN, USA, respectively) was used for cell lysis. Protein concentration was measured with Bradford assay (Sigma-Aldrich; Merck Millipore), and 10 μg from each sample was used for Western blotting. Polyvinylidene difluoride (PVDF) blots were incubated for 16 h at 4 °C with a primary antibody and 30 min at room temperature with the secondary antibody. The Restore Western Blot Stripping Buffer (Thermo Fisher Scientific Inc.) was used whenever PVDF blots stripping and reprobing was required. The Image J software, version 1.53f (National Institute of Health, Bethesda, MD, USA) was used for the densitometric analysis of the Western blotting bands.

The primary antibodies were specific for HLA-DR (dilution 1:100, cat. No 307648, Biolegend), CD80 (dilution 1:1000, cat. No. 15416, Cell Signaling Technology Inc., Danvers, MA, USA), intercellular adhesion molecule-1 (ICAM-1) (dilution 1:1000, cat. No. no 4915, Cell Signaling Technology Inc.), CD3-zeta (zeta) (dilution 1:100, cat. No. sc-20919, Santa Cruz Biotechnology, Santa Cruz, CA, USA), CD3-zeta phosphorylated on tyrosine 83 (p-zeta) (dilution 1:500, cat. No. ab68236, Abcam, Cambridge, UK), c-Myc (dilution 1:1000, cat. No. 9402, Cell Signaling Technology Inc.), marker of proliferation Ki-67 (Ki-67) (dilution 1:1000, cat no. NBP2-22112, Novus Biologicals, Abingdon, Oxon, UK), forkhead box P3 (FOXP3) (dilution 1:500, cat. No. 5298, Cell Signaling Technology Inc.), and β-actin (dilution 1:5000, cat. no 4967, Cell Signaling Technology Inc.). The anti-rabbit IgG-HRP-linked antibody (dilution 1:1000, cat. no 7074, Cell Signaling Technology Inc.) or the anti-mouse IgG-HRP-linked antibody (dilution 1:1000, cat. no. 7076, Cell Signaling Technology Inc.) were used as secondary antibodies. All original western blots are provided as a manuscript Appendix A.

### 4.3. Measurement of Monocyte Chemoattractant Protein-1

Monocyte chemoattractant protein-1 (MCP-1) was measured in the supernatants from isolated RPTEC cultures subjected or not to anoxia-reoxygenation. MCP-1 was measured on an EnSpire Multimode Plate Reader (Perkin Elmer, Waltham, MA, USA) with enzyme-linked immunosorbent assay using the LEGEND MAX Human MCP-1/CCL2 ELISA Kit (Biolegend). The minimum detectable MCP-1 concentration of the above kit is 1.6 pg/mL. Five such experiments were performed.

### 4.4. Assessment of Cell Proliferation

Cell proliferation was assessed after 7 days of RPTEC CD4+ T-cell co-culture in 96-well plates. Isolated untreated CD4+ T-cell cultures were used as control. Cell proliferation was measured by chemiluminescence on an EnSpire Multimode Plate Reader with the Cell Proliferation ELISA (Roche Diagnostics) using bromodeoxyuridine (BrdU) labeling and immunoenzymatic detection. Proliferation index was calculated for each of the four healthy volunteers that offered the CD4+ T-cells by the equation Proliferation index = optical density (OD) derived from RPTEC CD4+ T-cell co-culture: OD derived from the respective isolated CD4+ T-cell control culture. Four such experiments were performed.

### 4.5. Statistical Analysis

Statistical analysis was performed with the IBM SPSS Statistics for Windows, version 26 (IBM Corp., Armonk, NY, USA). In isolated RPTEC culture experiments, unpaired *t*-test was used for comparison of means. Since in RPTEC CD4+ T-cell co-culture experiments, each experiment is composed of distinguished RPTEC CD4+ T-cell couples, one-way repeated-measures analysis of variance (ANOVA) followed by the Bonferroni’s correction test was used. Results were presented as mean ± standard error of mean (SEM), and a *p* < 0.05 was considered statistically significant. After normalization for β-actin, for readers’ convenience, the Western blotting results were depicted after normalization for the control group.

## 5. Conclusions

Our study supports that RPTECs are subjected to direct allorecognition by CD4+ T-cells, indicating a broader and longer-lasting transfer of immunostimulatory cargo with the kidney graft than previously anticipated. Also, our data support that anoxia-reoxygenation increases the ability of RPTECs to stimulate directly alloreactive CD4+ T-cells, a fact that may enhance the probability for acute rejection at least during the short post-transplantation period.

## Figures and Tables

**Figure 1 ijms-22-01733-f001:**
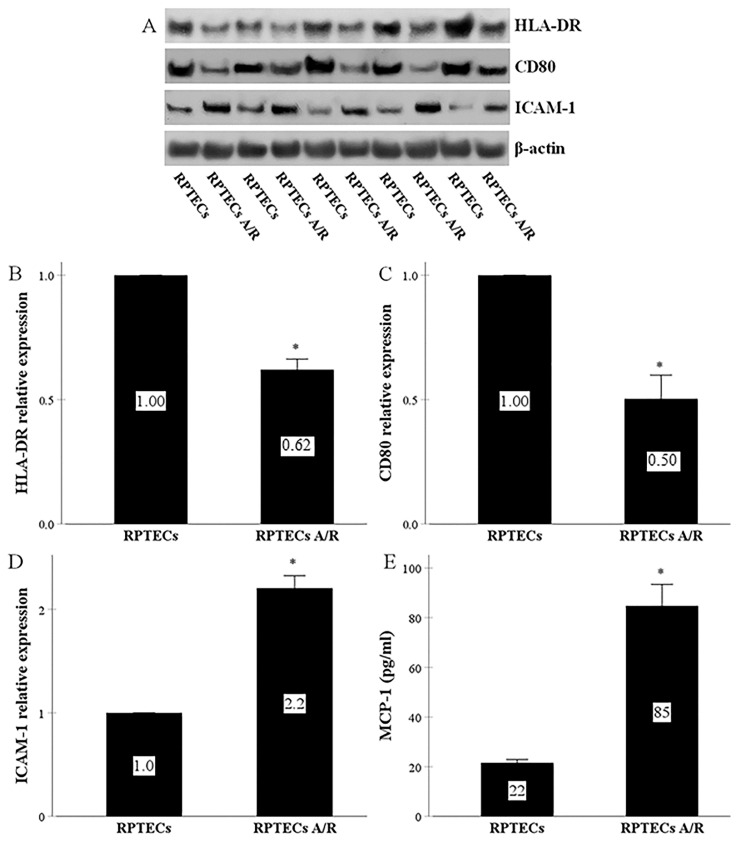
RPTECs express all the necessary molecules for direct allorecognition by CD4+ T-cells and produce MCP-1, whereas anoxia-reoxygenation affects all the above. RPTECs were cultured without any treatment or were subjected to 90 min of anoxia and 240 min of reoxygenation. Five experiments were performed. RPTECs expressed HLA-DR (**A**,**B**), CD80 (**A**,**C**), and ICAM-1 (**A**,**D**) and produced MCP-1 (**E**). Previous culture of RPTECs under anoxia-reoxygenation decreased HLA-DR (**A**,**B**) and CD80 (**A**,**C**), whereas it enhanced ICAM-1 (**A**,**D**) and MCP-1 (**E**). The Western blotting results were normalized for β-actin and then depicted after normalization for the control group. A/R stands for anoxia-reoxygenation, error bars correspond to SEM, and * indicates a *p* < 0.05.

**Figure 2 ijms-22-01733-f002:**
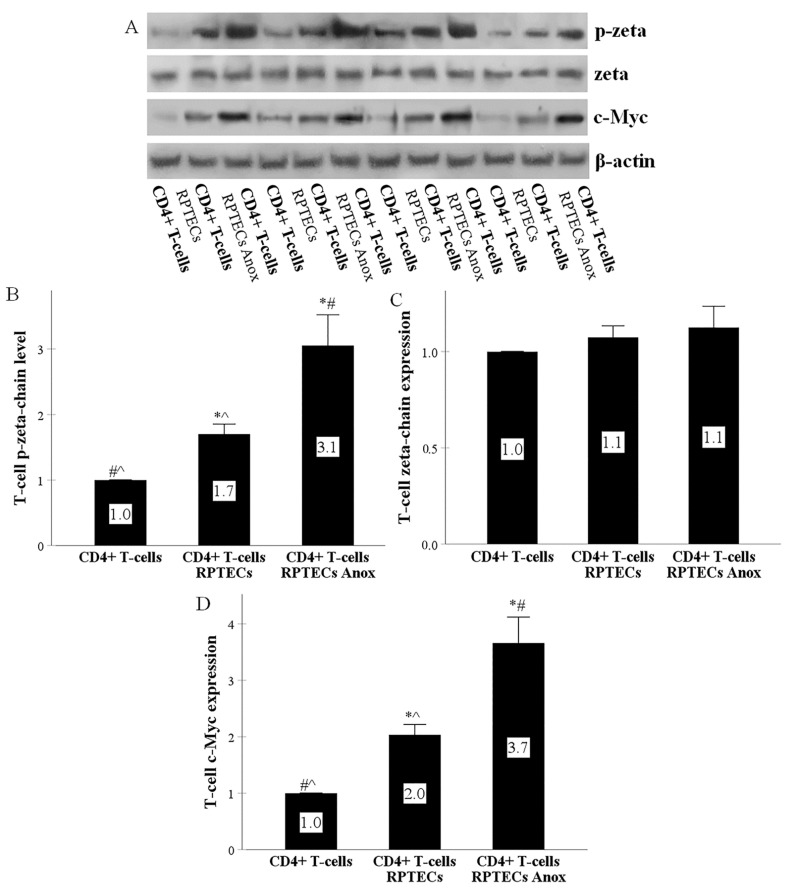
RPTECs activate the T-cell receptor and the co-stimulation signal transduction pathways in CD4+ T-cells, and anoxia-reoxygenation intensifies the above signal transduction pathways further. CD4+ T-cells were cultured alone or co-cultured with RPTECs. RPTECs were previously subjected or not to 90 min of anoxia. After 4 h of co-culture, CD4+ T-cells were collected. Four experiments were performed. Compared to isolated CD4+ T-cell culture, in CD4+ T-cell derived from RPTEC CD4+ T-cell co-culture, p-zeta-chain (**A**,**B**) and c-Myc (**A**,**D**) increased. Previous culture of RPTECs under anoxia induced further p-zeta (**A**,**B**) and c-Myc (**A**,**D**) enhancement in CD4+ T-cells. Zeta chain remained unaffected under all conditions (**A**,**C**). The Western blotting results were normalized for β-actin and then depicted after normalization for the control group. Error bars correspond to SEM. * indicates a *p* < 0.05 compared to isolated CD4+ T-cell culture, # a *p* < 0.05 compared to CD4+ collected from RPTEC CD4+ T-cell co-culture, and ^ a *p* < 0.05 compared to CD4+ collected from RPTEC CD4+ T-cell co-culture in which RPTECs were previously subjected to anoxia.

**Figure 3 ijms-22-01733-f003:**
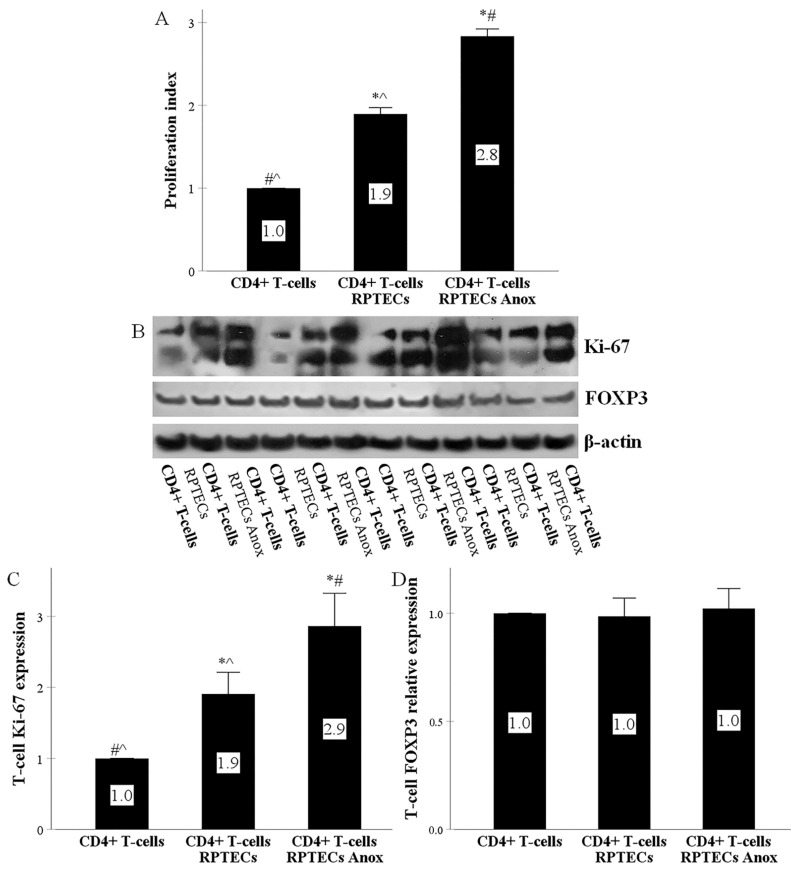
RPTECs trigger CD4+ T-cells to proliferate, anoxia-reoxygenation increases proliferation further, whereas CD4+ T-cell differentiation to regulatory T-cells does not occur. CD4+ T-cells were cultured alone or co-cultured with RPTECs for 7 days. RPTECs were previously subjected or not to 90 min of anoxia. Four such experiments were performed. Compared to isolated CD4+ T-cell culture, proliferation index was higher in RPTEC CD4+ T-cell co-culture and increased further in the case of previous RPTEC culture under anoxia. * indicates a *p* < 0.05 compared to isolated CD4+ T-cell culture, # a *p* < 0.05 compared to RPTEC CD4+ T-cell co-culture, and ^ a *p* < 0.05 compared to RPTEC (**A**). CD4+ T-cells were cultured alone or co-cultured with RPTECs. RPTECs were previously subjected or not to anoxia. After 3 days of co-culture, the CD4+ T-cells were collected. Four experiments were performed. Compared to isolated CD4+ T-cell culture, in CD4+ T-cell derived from RPTEC CD4+ T-cell co-culture Ki-67 (**B**,**C**) increased. Previous culture of RPTECs under anoxia induced further Ki-67 in CD4+ T-cells (**B**,**C**). FOXP3 expression in CD4+ T-cells remained unaffected under all conditions (**B**,**D**). The Western blotting results were normalized for β-actin and then depicted after normalization for the control group. Error bars correspond to SEM. * indicates a *p* < 0.05 compared to isolated CD4+ T-cell culture, # a *p* < 0.05 compared to CD4+ harvested from RPTEC CD4+ T-cell co-culture, and ^ a *p* < 0.05 compared to CD4+ collected from RPTEC CD4+ T-cell co-culture in which RPTECs were previously subjected to anoxia.

**Figure 4 ijms-22-01733-f004:**
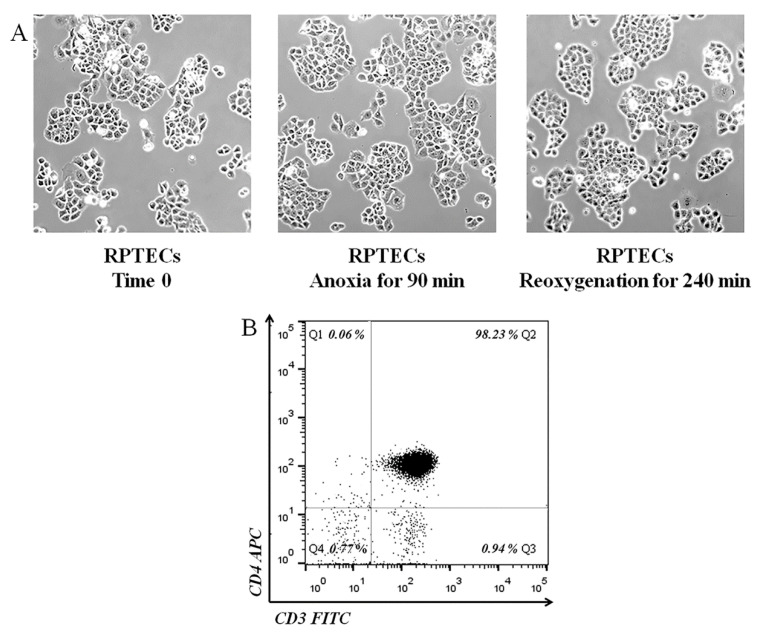
Assessment of the subjected to anoxia-reoxygenation RPTEC integrity and CD4+ T-cell purity. Cell imaging showed that after 90 min of anoxia and 240 min of reoxygenation, RPTECs retain their integrity. Magnification x100. (**A**). In CD4+ T-cell isolated from PBMCs, flow cytometry detected a purity higher than 98% (**B**).

## Data Availability

All original western blots are provided as a manuscript Appendix A. Any other analyzed datasets generated during the study are available from the corresponding author on reasonable request.

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
