# Peer review of "A Role for Human Renal Tubular Epithelial Cells in Direct Allo-Recognition by CD4+ T-Cells and the Effect of Ischemia-Reperfusion"

_ijms, 2021, doi:10.3390/ijms22041733_

Round 1
Reviewer 1 Report
This is a very well designed and well presented study that offers fresh thinking in the domain of allo-recognition in the context of IRI. The article presents the rationale, methodology and results of the study in a clear and comprehensive way. I applaud the authors for also presenting the limitations of the study in the discussion. I would recommend to add a couple of short paragraphs with their thoughts as to: a) what kind of translational/ pre-clinical and clinical studies they would recommend as the next step of their work and b) what kind of interventions they could recommend to down regulate the described allo-recognition.
Author Response
We would like to thank the reviewer for his/her positive comments and recommendations, which helped improve our manuscript.
- a) what kind of translational/ pre-clinical and clinical studies they would recommend as the next step of their work and b) what kind of interventions they could recommend to down regulate the described allo-recognition.
In the discussion section of the revised manuscript, just before the limitations, we added a paragraph and related references to express our opinion about possible interventions and useful future research.
“Our results indicate that kidney graft antigenicity remains high even after the first post-transplantation period since RPTECs are subjected to direct allorecognition. Thus, the search for the kidney transplantation holy grail, the graft tolerance, maybe better focused on the recipient’s immune system. For instance, the development of effective methods for enhancing specific to the graft Treg differentiation would be of great interest. Our data also indicate that at the immediate post-transplantation period, ischemia-reperfusion injury may increase the antigenicity of the kidney graft. Hence, efforts to alleviate kidney transplant ischemia-reperfusion injury by evolving and applying more efficient preservation methods may be proved beneficial. Finally, our study showed that ischemia-reperfusion increases RPTECs’ antigenicity by enhancing ICAM-1 expression. Research for blocking the interaction between cell adhesion molecules is underway, and the effect of such an approach on the field of kidney transplantation deserves evaluation”.
Reviewer 2 Report
The present study by Eleftheriadis and coworkers examined the role of human RPTECs in mediating direct allorecognition by CD4+ T-cells. It also examined the potential role of anoxia- reoxygenation in enhancing RPTECs ability to stimulate alloreacyive CD4+ T cells.
1- Author needs to discuss the clinical impact of the current study; how could these findings benefit patients.
2- Are alternative molecules coordinately upregulated such as VCAM-1, CD40 which are involved in T- cell activation.
3- The data presented only as protein expression. Does RNA level confirms unregulated expression of the in vitro cultured cells.
4-The introduction would be strengthened by a discussion of the mechanism by which ischemia-reperfusion injury of the kidney graft mediate subsequent acute rejection.
5- Author needs to indicate in figure caption or the y axis of figure 2,3,4 that protein levels were normalize to the B-actin.
6- Manuscript should be reviewed for grammar/ syntax errors.
7- Figure caption need to be expanded to include informative description of each of the data in the figure e.g. A,B,C,D.
Author Response
We would like to thank the reviewer since his/her valuable comments helped us on improving our manuscript.
1- Author needs to discuss the clinical impact of the current study; how could these findings benefit patients.
In the discussion section of the revised manuscript, just before the limitations, we added a paragraph and related references to express our opinion about possible interventions and useful future research.
“Our results indicate that kidney graft antigenicity remains high even after the first post-transplantation period since RPTECs are subjected to direct allorecognition. Thus, the search for the kidney transplantation holy grail, the graft tolerance, maybe better focused on the recipient’s immune system. For instance, the development of effective methods for enhancing specific to the graft Treg differentiation would be of great interest. Our data also indicate that at the immediate post-transplant period, ischemia-reperfusion injury may increase the antigenicity of the kidney graft. Hence, efforts to alleviate kidney transplant ischemia-reperfusion injury by evolving and applying more efficient preservation methods may be proved beneficial. Finally, our study showed that ischemia-reperfusion increases RPTECs’ antigenicity by enhancing ICAM-1 expression. Research for blocking the interaction between cell adhesion molecules is underway, and the effect of such an approach on the field of kidney transplantation deserves evaluation”.
2- Are alternative molecules coordinately upregulated such as VCAM-1, CD40 which are involved in T- cell activation.
Of course, other molecules are also involved in T-cell activation. We decided to evaluate HLA-DR as a representative of MHCII molecules, CD80, the archetype co-stimulatory molecule, and from the cell adhesion molecules the ICAM-1, which plays a pivotal role in the immunological synapse formation (Dustin M, Immunity 30: 482-492, 2009).
3- The data presented only as protein expression. Does RNA level confirms unregulated expression of the in vitro cultured cells.
Although WB is a relatively rigorous method, it is still invaluable, and in any issue of high impact journals, such as the Science, someone can find studies that use this method. Contrary to PCR, WB detects the final expression of a protein regardless of any posttranslational degradation. The best example is the HIF1α protein, which upon anoxia is upregulated not due to increased gene transcription but because of inhibition of its degradation. Under certain circumstances, zeta-chain, which was evaluated in our study, is also subjected to degradation (Eleftheriadis et al., DOI: 10.2174/157436206777012039). More importantly, contrary to PCR, WB can detect posttranslational protein modifications. Assessment of phosphorylated zeta-chain was pivotal for our study.
4-The introduction would be strengthened by a discussion of the mechanism by which ischemia-reperfusion injury of the kidney graft mediate subsequent acute rejection.
This was one of the subjects of our study. In the introduction of the revised manuscript, we discussed the suggested mechanism by which ischemia-reperfusion injury of the kidney graft mediate subsequent acute rejection. In the second paragraph of the introduction, we wrote, “It has been suggested that kidney transplant ischemia-reperfusion injury induces cell necrosis and release of damage-associated molecular patterns. The latter are recognized by specific receptors, such as the Toll-like receptors, leading to activation of innate immune cells and the production of cytokines and chemokines. This inflammatory milieu induces further renal cell necrosis through necroptosis and in parallel recruits and activates cells of the adaptive immune system facilitating graft rejection”.
5- Author needs to indicate in figure caption or the y axis of figure 2,3,4 that protein levels were normalize to the B-actin.
In the revised manuscript, in the last sentence of subsection statistical analysis, we wrote: “For reader’s convenience, after normalization for β-actin, the western blotting results were depicted after normalization for the control group”. Also, we wrote the same in the caption of any figure with WB.
6- Manuscript should be reviewed for grammar/ syntax errors.
We re-check the manuscript for correcting grammar/sytax errors.
7- Figure caption need to be expanded to include informative description of each of the data in the figure e.g. A,B,C,D.
In the revised manuscript, we expanded figure captions as recommended. Actually, some problems occurred during the editing by the journal. We corrected them too.
Reviewer 3 Report
Dear Authors,
Thank you for the opportunity to review your paper entitled A role for human renal tubular epithelial cells in direct allorecognition by CD4+ T-cells and the effect of ischemia-reperfusion.
The study discusses an important matter in the nephrological area, however, some major improvements are needed to consider the manuscript suitable to be published in IJMS.
1) There is no basic clinical characteristic of the four subjects involved in the study. It is of great interest how were their creatinine levels, WBC count and the percentage of lymphocytes in the blood analysis. BMI, as well as MDRD, would be also appreciated.
2) How many WBs have been done? Please keep in mind that the paper is based on western data, so at least three repetitions should be done.
3) I am not sure if the loading control for both Fig 2 and Fig 3 are from the same gel run. The bands look different when analyzed with binary masking. Please explain and replace if needed with the corresponding Actin bands.
4) I am not sure why the Authors did not perform PCR on the cells. Please keep in mind that considering only WB expression data might be terribly misleading. On the other hand, not sure if the timing will allow observing any changes in molecular patterns. What if the donors' variability highly impacted the results?
5) It would be great to re-confirm your findings using basic immunofluorescence staining for confocal study of your cells using abs against ICAM-1; MCP-1 etc...
6) Please insert correct symbol for Celcius degree (not the dot in mid-space)
7) In the discussion section I would like to see discussed below-mentioned papers/issues that will significantly broaden the audience of your pape:
- CD80 and CTLA-4 as diagnostic and prognostic markers in adult-onset minimal change disease (doi: 10.7717/peerj.5400)
- HLA-Class II Artificial Antigen Presenting Cells in CD4+ T Cell-Based Immunotherapy (https://doi.org/10.3389/fimmu.2019.01081)
- ICAM/VCAM in kidney disease and their link to CVD and mortality rate:https://doi.org/10.1186/s12882-017-0457-1
8) Please carefully discuss the limitations of your study.
I would be more than happy to see the revised version of your manuscipt.
Best,
Author Response
We would like to thank the reviewer since his/her comments helped us clarify certain points and improve our manuscript.
1) There is no basic clinical characteristic of the four subjects involved in the study. It is of great interest how were their creatinine levels, WBC count and the percentage of lymphocytes in the blood analysis. BMI, as well as MDRD, would be also appreciated.
In the revised manuscript, we clarified this point better. We wrote, “All participants were personnel of our clinic-laboratory. Their medical records were evaluated, and they were subjected to physical examination and routine laboratory tests. Their BMI was within the normal range, and eGFR calculated by the MDRD equation was higher than 90 ml/min/1.73 m2”.
2) How many WBs have been done? Please keep in mind that the paper is based on western data, so at least three repetitions should be done.
As it is written in the first paragraph of subsection 2.2 of the manuscript, five experiments were performed for assessing the proteins of interest in RPTECs subjected or not to I-R, and four experiments for evaluating the proteins of interest in CD4+ T-cells isolated for RPTEC CD4+T-cell co-culture. We also noted the number of the experiments in the figure captions of the revised manuscript. All the WB experiments are depicted.
3) I am not sure if the loading control for both Fig 2 and Fig 3 are from the same gel run. The bands look different when analyzed with binary masking. Please explain and replace if needed with the corresponding Actin bands.
Each actin corresponds to the depicted proteins of each figure since they are derived from the same gel run. All gels were blotted with the corresponding antibodies, striped and reblotted. Any discrepancy in intensity or marginal misalignment is attributed to the exposure, different band size, serial stripping-reblotting, the strength of the antibody (especially Cell Signaling’s actin antibody), and correspondence in the final figure. All original WBs, unedited, are uploaded to the journal as per MDPI’s policy. As written in the last sentence of subsection statistical analysis “After normalization for β-actin, for readers' convenience, the western blotting results were depicted after normalization for the control group.” Also, we wrote the same in the caption of any figure with WB.
4) I am not sure why the Authors did not perform PCR on the cells. Please keep in mind that considering only WB expression data might be terribly misleading. On the other hand, not sure if the timing will allow observing any changes in molecular patterns. What if the donors' variability highly impacted the results?
Although WB is a relatively rigorous method, it is still invaluable, and in any issue of high impact journals, such as the Science, someone can find studies that use this method. Contrary to PCR, WB detects the final expression of a protein regardless of any posttranslational degradation. The best example is the HIF1α protein, which upon anoxia is upregulated not due to increased gene transcription but because of inhibition of its degradation. Zeta-chain, which was evaluated in our study, and under certain circumstances, it is also subjected to degradation (Eleftheriadis et al., DOI: 10.2174/157436206777012039). More importantly, contrary to PCR, WB can detect posttranslational protein modifications. Assessment of phosphorylated zeta-chain was pivotal for our study.
Actually, in the time points used in our study, we detected changes. As it can be seen by our previous publications, we evaluated such proteins in cell cultures using similar time points. Besides, many other studies demonstrated such changes even in shorter periods. For instance, after T-cell activation, WB detected zeta chain phosphorylation as early as 3 minutes (Oers et al., MOLECULAR AND CELLULAR BIOLOGY, Sept. 1993, p. 5771-5780, Figure 3A). As another example, c-myc mRNA has been found upregulated after 1 hour of T-cell stimulation (Lindsten et al., The EMBO Journal vol.7 no.9 pp.2787-2794, 1988).
We used four different healthy volunteers to obtain CD4+ T-cells. In order to obtain reliable results, in each of the four RPTEC CD4+ T-cell co-culture experiments, CD4+ T-cells from the same individual were used. In the revised manuscript, we clarified this point better (Last sentence of subsection 2.1).
5) It would be great to re-confirm your findings using basic immunofluorescence staining for confocal study of your cells using abs against ICAM-1; MCP-1 etc...
Unfortunately, we do not have confocal microscopy in our facility. However, as regards to ICAM-1 we used WB, which is as semiquantitative as confocal microscopy is. Regarding MCP-1, a chemokine produced and secreted by the cells, we used ELISA, which has higher sensitivity and specificity than confocal microscopy.
6) Please insert correct symbol for Celcius degree (not the dot in mid-space)
We corrected the symbol for Celciuc degree.
7) In the discussion section I would like to see discussed below-mentioned papers/issues that will significantly broaden the audience of your pape:
- CD80 and CTLA-4 as diagnostic and prognostic markers in adult-onset minimal change disease (doi: 10.7717/peerj.5400)
- HLA-Class II Artificial Antigen Presenting Cells in CD4+ T Cell-Based Immunotherapy (https://doi.org/10.3389/fimmu.2019.01081)
- ICAM/VCAM in kidney disease and their link to CVD and mortality rate:https://doi.org/10.1186/s12882-017-0457-1
We incorporated all the suggested citations in the revised manuscript. Ref 24, 36, and 37 of the revised manuscript, and we thank the reviewer for his excellent point and his effort.
8) Please carefully discuss the limitations of your study.
We have discussed the limitations of our study in two paragraphs, preceding our manuscript's last paragraph with the conclusions.
Round 2
Reviewer 3 Report
Dear Authors,
I am happy with your response as well as changes that had been made to the draft. I would like to suggest the acceptance the paper.
On the other note. Just FYI: I am aware that westerns are broadly used in almost every single mechanistic paper and my comments were not against you - they were just to stimulate your efforts and make your future papers more influential. Last year I reviewed over 10 papers for journals with IF>10 and all of them contained western data - you are right. But every single paper presented also second "wave" that confirmed western data. Our goal is to make science better - so please take my advice as friendly reminder and include this point in your future research.
best!